# Assessing Femoral Head Medialization in Developmental Hip Dysplasia Type 1 and Type 2 Hip Separation

**DOI:** 10.3390/diagnostics14202317

**Published:** 2024-10-18

**Authors:** Sonay Aydin, Onder Durmaz, Erdem Fatihoglu, Ozlem Kadirhan, Erdal Karavas

**Affiliations:** 1Department of Radiology, Faculty of Medicine, Erzincan University, 24100 Erzincan, Turkey; sonay.aydin@erzincan.edu.tr (S.A.); erdemfatihoglu@hotmail.com (E.F.); ozlemkkadirhan@gmail.com (O.K.); 2Department of Radiology, Faculty of Medicine, Bandirma 17 Eylül University, 10200 Balikesir, Turkey; erdalkaravas@hotmail.com

**Keywords:** Graf classification, developmental dysplasia, hip, femur, measurement

## Abstract

Background/Objectives: The prevalence of developmental hip dysplasia is estimated to be 0.1–2 per 1000 infants. Hip imaging by ultrasonography is considered to be the gold standard method for screening and detecting developmental dysplasia of the hip (DDH), as per the Graf categorization. The classification of hip differentiation into type 1 and type 2 is determined by the alpha angle, as assessed by the Graf classification. Type 1 hips are defined as those with an alpha angle exceeding 60 degrees, whilst type 2 hips are defined as those with measurements falling within the range of 50 to 59 degrees. Methods: The computerized patient card in our institution had a compilation of 208 hip photographs taken from 110 patients, with 98 of them being bilateral. The acquisition of these photos occurred from January 2020 to December 2020. A retrospective review was conducted on the ultrasound (US) scans, with a specific emphasis on the outcomes related to type 1 and type 2 hips. Results: There were 108 high-resolution US photos in the type 1 hip group and 100 high-resolution US images in the type 2 hip group. In terms of unilateral or bilateral cases, gender, or age, no statistically significant differences were seen between the two groups (*p* > 0.05). The FMD model exhibited a sensitivity of 86% and specificity of 70% in effectively predicting the presence of type 1 mature hips when the values surpassed 2.9 mm. The AUC (area under the curve) value achieved was 0.628. Conclusions: The process of diagnostic categorization may occasionally encounter challenges in accurately differentiating between type 1 and type 2 hip separation subsequent to a hip ultrasound examination. The findings of our analysis indicate that the assessment of the FMD is a highly successful method, demonstrating both high specificity and sensitivity in differentiating between various scenarios.

## 1. Introduction

Patients with developmental dysplasia of the hip (DDH) exhibit abnormalities in the bone and soft tissues of the hip joint, demonstrating detrimental impacts on the acetabulum, femoral head and neck, and pelvis due to musculoskeletal alterations, subluxation, or total dislocation. The hip joint often assumes the shape of a deep acetabulum throughout a patient’s intrauterine life, becoming shallow at birth and deepening thereafter as it matures, ultimately enveloping the femoral head entirely. In cases of DDH, the acetabulum remains shallow, and the femoral head develops in an aberrant, non-anatomical position [1,2].

Developmental dysplasia of the hip encompasses a broad spectrum of pathologies, ranging from transient immaturity of the hip to total dislocation of the hip joint. This disorder may manifest whilst the patient is intrauterine and can also arise post-childbirth. The incidence varies from 0.07 to 77 per 1000 live births, demonstrating considerable variation across various racial groups and geographical areas. If untreated, DDH is associated with long-term problems that can lead to considerable disability in early adulthood [3,4].

There are several recognized risk factors for DHD. These risk factors may include a positive family history, breech presentation before birth, and abnormalities such as neonatal clubfoot or torticollis [3,5,6,7].

Clinical hip instability in neonates was initially documented in 1879. A clinical assessment for hip instability was proposed by Le Damany and Safety in 1910. In 1937, they garnered further attention through Ortolani. The technical analysis and conclusions regarding dislocation or subluxation were presented by Palmen in 1961 and by Barlow in 1962 [8,9,10,11].

In early infancy, the mostly cartilaginous hip joint and the unossified femoral head render radiography an inadequate and ineffective assessment. The images are often captured with the hips in a neutral position and in flexion–abduction (frog leg), which may not accurately depict the hip joint in displacement. After six months of age, radiography emerges as a proficient diagnostic modality owing to the ossification of the femoral head nucleus. An anteroposterior picture of the hips in a neutral position is routinely acquired and evaluated during DHD screening. Assessing the relationship among the radiolucent femoral head, bone metaphysis, and acetabulum is crucial [12,13,14].

Hip ultrasonography is frequently utilized as a diagnostic and screening method for developmental dysplasia of the hip (DDH). Ultrasound examination can be safely employed in the pediatric population due to its non-ionizing characteristics, which mitigate the risk of radiation exposure. Nonetheless, attaining standards might be arduous due to its dependence on the operator’s expertise and reliability. Due to the cartilaginous anatomy of the hip joint in the newborn era, its components are readily visualized by ultrasound. Nevertheless, meticulous oversight is necessary, and stringent protocols are established [15,16,17].

In the Graf classification, the utilization of US as a hip imaging technique for the purpose of screening and diagnosing DDH is considered to be the most reliable and widely accepted method. This technique categorizes patients into four primary groups and a further division of the classification is clearly outlined in Table 1 [18].

In several industrialized nations, the utilization of US scanning is limited to specific instances, but in our country, it is primarily employed for the goal of screening. US can be readily conducted until the 6th month following childbirth, but pelvic radiography is employed after the 6th month. Differentiating between type 1 mature and type 2 physiological immature hips in early US scans can be challenging because of the reliance on the user’s interpretation. Therefore, clarifying the withdrawal of patients from follow-up can be challenging at times [19].

This study aims to enhance diagnostic accuracy using a newly developed measurement approach known as the femoral head medialization diameter (FMD) method in cases of hip dysplasia where the Graf method yields equivocal diagnoses. Consequently, we believe that this technique (FMD) will serve as an alternate diagnostic approach to distinguish between Graf type 1 and type 2 hips and may also enhance the Graf method in determining the continuation or cessation of treatment.

## 2. Materials and Methods

The study received approval from the ethical committee of the center where it was carried out. The study center where the approval of the ethics committee was obtained is Erzincan University Faculty of Medicine (Erzincan/Turkey). Informed consent was obtained from the patients’ parents involved in the study. Written informed consent has been obtained from the patients’ parents to publish this paper (the ethics committee number is 34364579-714.01.02-E.40236, and the date of approval was 5 February 2021). The study was conducted at Erzincan University.

A retrospective re-examination was conducted on a total of 208 hip ultrasound images obtained from 110 patients (98 of whom had bilateral findings) aged between 6 and 8 weeks. These images, categorized as type 1 and type 2, were collected from the computerized patient card in our hospital between January 2020 and December 2020. All 98 infants who participated in the study were healthy and born at full term. Patients who underwent hip surgery for various reasons, those who had congenital diseases accompanying hip dysplasia, and premature infants were not included in the study. Furthermore, retrospective images that did not satisfy the relevant criteria were also omitted.

The patients underwent US evaluations using the Aplio 500 ultrasound machine (Toshiba Medical Systems, Tokyo, Japan), equipped with a 7–9 MHz linear probe. The infants did not receive any sedative medication throughout the examination. In the hip US examination method, infants are placed in a lateral decubitus posture and their legs are brought to a neutral position, with the knees and hips slightly flexed. After applying gel to the high-frequency (linear) probe, repeated measurements are taken from the appropriate images obtained. The coronal sections used in the study were obtained from all available images. The examinations were concluded by consensus by two radiologists, one with 8 years of expertise and the other with 15 years of experience.

The recently devised technique involved measuring the distance between the central femoral head and a horizontal line that passes through the iliac wing. This measurement is referred to as the FMD value (Figure 1). The determination of the center of the femoral head in hip US pictures was achieved by calculating the average of the longest vertical diameter of the femoral head, as measured in the coronal plane (Figure 2).

The parameters included in the study were age, gender, FMD measurement values, and ultrasound images showing type 1 and type 2 hips.

### Statistical Analysis

The study data were analyzed using IBM SPSS Inc.‘s Statistical Package for the Social Sciences (SPSS) version 20 for Windows 11 (Chicago, IL, USA). The data were verified to follow a normal distribution using the Kolmogorov–Smirnov test. Numerical data that follow a normal distribution are typically represented as the mean plus or minus the standard deviation. The difference in FMD values between the type 1 and type 2 hip groups, as well as between genders, was assessed using a Student’s *t* test.

The gender distribution among the groups was analyzed using a chi-square test. The relationship between the FMD value and age was assessed using a Pearson correlation analysis. The predictive accuracy of FMD readings in determining type 1 hips was evaluated using a ROC analysis.

A *p*-value less than 0.05 was deemed to be statistically significant.

## 3. Results

The study included 108 images of the type 1 hip group and 100 images of the type 2 hip group. Out of a total of 208 hips, 57% (*n* = 120) were from female newborns and 43% (*n* = 88) were from male infants. Out of the 108 type 1 hips, 55% were female newborns (60 hips) and 45% were male infants (48). Out of the 100 type 2 hips, 60% were female newborns (60 hips) and 40% were male infants (40 hips). There was no statistically significant difference in terms of gender between the two groups (*p* > 0.05). The average ages of type 1 and type 2 hips were computed as 63 and 60 days, respectively. There was no statistically significant difference in age between the two groups (*p* > 0.05).

A significant and positive correlation was found between age and FMD values (r = 0.87, *p* = 0.01). No significant correlation was found between gender and FMD values (*p* = 0.07).

Mean FMD measurements were significantly higher in the type 1 group compared to the type 2 group (3.4 +/− 0.7 mm vs. 2.2 +/− 1.1, *p* < 0.01). Using an FMD cut-off value of higher than 2.9 mm shows an 86% sensitivity and a 70% specificity in the diagnosis of a type 1 hip. The area under the curve (AUC) was calculated as 0.628 and the *p*-value was 0.02.

## 4. Discussion

Developmental dysplasia of the hip (DDH) is a significant global clinical issue that impacts infants and their families. DDH is the predominant hip disorder in children and can lead to significant and lasting alterations if not addressed [20]. Unstable dislocation of the hips may go unnoticed if screening is solely conducted through physical examination. Furthermore, relying just on a physical examination may result in inaccurate positive results.

Ultrasound examination, popularized and standardized by Graf in the 1980s, is very important in this respect. Accurately differentiating between type 1 and type 2 hips using the Graf classification approach is crucial for predicting outcomes and monitoring patient progression [6,18,21]. After a detailed literature study, we thought that an alternative imaging technique may be useful to more accurately recognize and follow up hip dysplasia, which can cause permanent disability in the community. Therefore, our study aimed to use the FMD, a new measurement that could potentially help to differentiate between type 1 and type 2 hips.

Acetabular dysplasia and ligamentous laxity have been reported in a number of publications as the main factors contributing to the development of DDH. Maturation of the hip joint, which is an ever-changing process, occurs both during fetal development and after birth. Previous studies have shown that neonates with clinically unstable hips, as detected by ultrasound, may recover spontaneously without any intervention. Shitrit et al. showed that only 10% of sonographically diseased hips in 1- and 3-day-old infants continue to show abnormalities until the 6th month of life. Although the incidence of this event is very rare, it is important to note that removal of a pathological hip from follow-up may have significant long-term consequences for the patient [22,23].

The bone roof is the primary determinant of the alpha angle. The cartilage roof is the defining factor of the beta angle. Nevertheless, according to the Graf approach, the primary angle in hip typing is referred to as the ‘alpha angle’. However, it is well-established that the beta angle, also known as the cartilage roof, and the age of the patient are significant factors in subtyping. In the context of case follow-up and prognosis, it is imperative to consider the combined assessment of both angles and the age at which they are situated in order to arrive at an informed choice [18,22,24]. When the alpha angle is measured at degrees of 60 and above, it is classified as a type 1 (mature) hip and cases are excluded from follow-up. In the first 6 months after birth, when the alpha angle is measured at 50–59°, a type 2 hip is diagnosed. In the initial three months, hip measurements ranging from 54° to 59° are categorized as type 2a(+), whereas readings falling within the range of 50–54° are designated as type 2a(−) hips. Infants older than 3 months with an alpha angle ranging from 50–59° are categorized as having a type 2b hip. If the alpha angle is measured between 43° and 49° (assuming the beta angle is less than 77°), it is classified as type 2c [18,22].

There are opinions suggesting that there is a small delay in the maturation process of the acetabular roof in type 2a(+) hip cases. According to the same study, infants with type 2a(−) hips experience a significant delay in this developmental process. According to a study in the literature, approximately 10 per cent of patients diagnosed with a type 2a hip may develop a dysplastic hip. In another study, abnormal hip development was documented in 4.7% of individuals diagnosed with a type 2a(+) hip and 14.6% of individuals diagnosed with a type 2a(−) hip [18,20,25].

Acetabular dysplasia is characterized by the presence of type 2b hips. Close surveillance and medical intervention from radiology and orthopedic clinics may be necessary. Type 2c hips are commonly referred to as dysplastic hips. Prompt initiation of treatment is imperative, irrespective of an individual’s age. At this stage, a range of orthopedic interventions and vigilant supervision are necessary [15,16,18,22,25].

The cases included in our study were categorized into type 1 and type 2 hips based on FMD measures, without any subclassification based on hip type.

As a result of our detailed literature search, it is apparent that there has been no previous study on the measurement of the medialization of the femoral head. However, Kosar et al. performed a remarkable similar study evaluating the medialization of the acetabular roof. The aim of this study by Kosar et al. was to evaluate the correlation between acetabular roof medialization and the prognosis of type 3 and type 4 hip treatment. A total of 35 hips were evaluated in this study. As a result of their study, they reported a strong correlation between acetabular roof medialization and prognosis [16]. In our study, FMD measurement was used to differentiate between type 1 and type 2 hips and a statistically significant difference was found. Future research may differentiate between type 1 and type 2 hips using acetabular roof medialization. Furthermore, given the findings of Kosar et al. [15] on the advantageous effects of acetabular roof medialization measurement for treatment, future studies on type 2 hip diseases may yield more reliable results. Further investigation of the correlation between alpha and beta angles and the FMD in future studies may provide more reliable findings. In our study group, in a 60-day-old infant diagnosed with an immature hip by the Graf method, the FMD medial values showed results supporting this diagnosis (Figure 3).

Furkatovich and colleagues reported that sonographic examination is the superior option to X-ray for assessing hip disease in neonates. They proposed that radiation-free sonographic care is superior for early diagnosis. They also indicated that cartilage and adipose tissue can be more effectively assessed using sonographic evaluation in the early years of life, making it the most current and unequivocal diagnostic instrument for developmental hip dislocation. An advantageous alternative for diagnosis is the ultrasound examination of the hip joints, facilitated by the latest advancements in diagnostic equipment. We assert that the FMD technique, informed by these data and findings, and utilized with sonographic imaging, will be beneficial in the diagnosis and early intervention of developmental hip dysplasia [26].

The 2022 study by Han, J. and Li, Y. showed that universal ultrasound screening significantly reduced the incidence of late-detected cases compared with alternative screening programs; however, it may lead to overtreatment and significant treatment costs due to misinterpretation. The authors suggested that the timing of ultrasonography and other imaging should be appropriate for effective treatment planning, and that the long-term operational costs of GKD screening are more cost-effective compared to surgical and non-surgical interventions. Translated with DeepL.com (free version) [27].

A recent study by Gyurkovits and colleagues indicated that infants with developmental hip dysplasia should be assessed using clinical and sonographic criteria, and the incorporation of supplementary imaging modalities will facilitate accurate diagnosis and prompt treatment in some cases [28].

The significance of DDH screening is acknowledged; however, the precise technique and approaches, including whether it should be universal or selective, remain undetermined. Considering that DDH may result in a permanent handicap, the advantageous impacts of universal ultrasound have been acknowledged, and its adoption was suggested over fifty years ago. Furthermore, it is acknowledged that supplementary imaging and vigilant monitoring are required in certain cases that remain unresolved [29,30]. This section introduces different diagnostic approaches that enhance the diagnosis of hip dysplasia and facilitate early intervention. Given the straightforward usage of the FMD and its applicability with the graph approach, we propose that it may serve as a diagnostic tool for hip dysplasia cases.

In a study conducted by Hareendranathan et al. in 2017, they presented 3D US image analysis as an additional diagnostic method to the graphical method, combining the discrimination of normal and hip dysplasia cases, semi-automatic segmentation of the acetabular surface, automatic calculation of shape features, and an automatic classifier. They even reported that this technique is promising and needs larger-scale clinical studies [31]. In this original study, we aimed to recognize and follow up an important musculoskeletal disease that results in high costs in terms of outcomes in the society by using the FMD technique, which has not been used anywhere so far, and sonography, which is a cheap imaging method available everywhere.

The FMD value described in our study can be determined by taking two measurements on a single ultrasound image during the same hip examination. This easy measurement method will eliminate the confusion in differentiating between type 1 and type 2 hips. Measurement in this respect can prevent unnecessary follow-up and misdiagnosis of patients. In this way, it can add practicality to intensive ultrasound examinations.

Our study has some limitations. Due to the retrospective character of our study, the reliability of the measurement may be reduced compared to real-time images. Furthermore, the population size in our study group is still insufficient and studies with larger populations are needed. In addition, our other limitation is that the moderate AUC values limit the use of results in daily practice. In this study, patients were only divided into two main groups, those with type 1 and type 2 hips. Including the type 2 hip subclassification and providing mean FMD values for each subclass would have increased the depth and comprehensiveness of the study. Although this is accepted as a limitation, it can be accepted that it will add strength to the study considering that the results can be extended to all type 2 hips. Finally, due to the lack of treatment data, it was not possible to obtain information about the comorbid conditions and final prognosis of the patients.

## 5. Conclusions

We know that congenital hip dysplasia in babies triggers some functional musculoskeletal diseases in their later life and causes morbidity. In most countries, especially in our country, hip ultrasound scans are performed to prevent this condition. The recently developed femoral medialization value assessment has the potential to be a reliable tool in the identification of type 1 hips. The FMD value must exceed 2.9 mm on sonographic examination to diagnose a type 1 mature hip and to terminate follow-up. Based on the findings of our study, hips with dimensions of less than 2.9 mm should be classified as immature. In conclusion, the measurement of FMD facilitates the differentiation between type 1 and type 2 hips and provides a more reliable discharge of patients from follow-up.

## Figures and Tables

**Figure 1 diagnostics-14-02317-f001:**
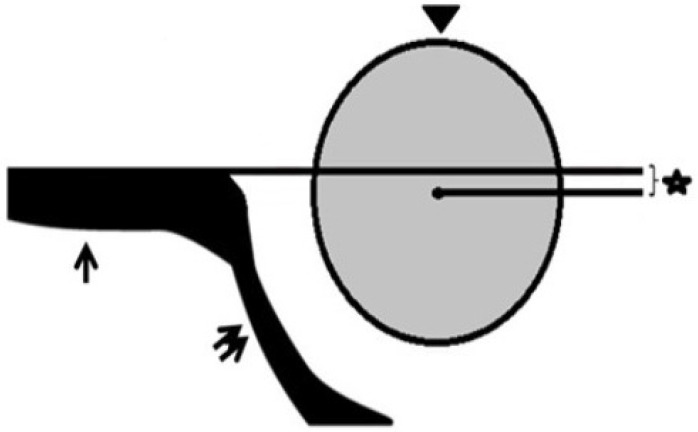
The iliac bone (single arrow), iliac roof (double arrow), and femoral head (arrow head) demonstrated in picture form. The distance indicated by the star is the measurement between the femoral decapitation center and the horizontal line passing through the iliac bone.

**Figure 2 diagnostics-14-02317-f002:**
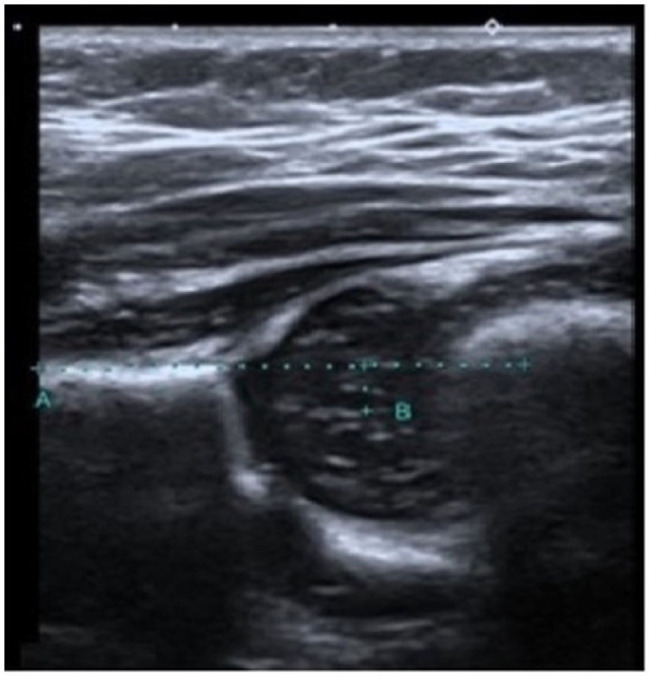
The FMD value is measured as 3.3 mm in a 62-day-old type 1 mature hip with an alpha angle of 67 and a beta angle of 53 (FMD refers to the distance that remains between point B and the decussated A line, which is drawn parallel to the femur from the iliac bone. Point B refers to the center of the femoral head).

**Figure 3 diagnostics-14-02317-f003:**
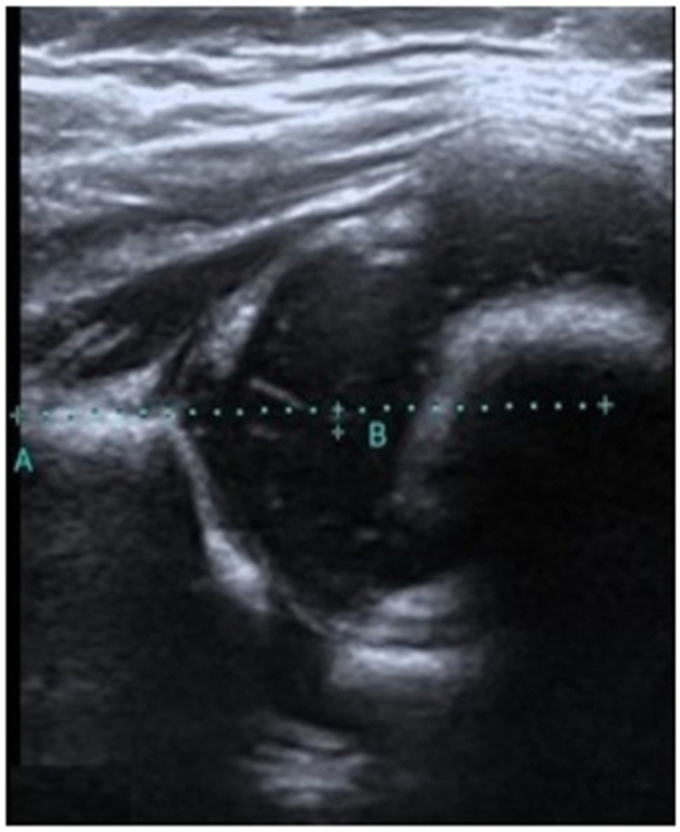
Alpha angle of 58 and beta angle of 55 measured in a 60-day-old type 2 physiological immature hip; the FMD value measured 2.1 mm (FMD refers to the distance that remains between point B and the decussated A line, which is drawn parallel to the femur from the iliac bone. Point B refers to the center of the femoral head).

**Table 1 diagnostics-14-02317-t001:** Graf classification chart image used to detect congenital hip dysplasia and normal hip measurements.

Type	Maturity	Bony Roof	Bony Angle	Bony Rim	Cartilage Roof	β Angle	Age
Type l	Mature	Good	α ≥ 60°	Sharp	Good coverage femoral head	$Ia=\beta <55^{\circ }$ $Ib=\beta >55^{\circ }$	All
Type 2a+	Immature but appropriate for age	Adequate	50–59°	Blunt	Coverage femoral head		<3 mo
Type 2a−	Immature and inappropriate for age	Deficient	50–59°	Rounded	Coverage femoral head		<3 mo
Type 2b	Delay in development	Deficient	50–59°	Rounded	Coverage femoral head		>3 mo
Type 2c	Stable or unstable	Severely deficient	43–49°	Rounded/flat	Still coverage femoral head	β < 77°	All
Type D	Decentered hip	Severely deficient	43–49°	Rounded/flat	Displaced	β > 77°	All
Type 3	Eccentric hip	Poor	<43°	Flat	Labrum pressed upwards		All
Type 4	Eccentric hip	Poor	<43°	Flat	Labrum pressed downwards		All

## Data Availability

The original contributions presented in the study are included in the article, further inquiries can be directed to the corresponding author.

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
