# Peer review of "Assessing Femoral Head Medialization in Developmental Hip Dysplasia Type 1 and Type 2 Hip Separation"

_diagnostics, 2024, doi:10.3390/diagnostics14202317_

Round 1
Reviewer 1 Report
Comments and Suggestions for Authors
Thank you for given me opportunity to review this interesting manuscript. From my perspective introduction is too long for readers who are not novice to this topic. It is suggested to describe the problem in brief manner and explain purposes of this study. In method section and further in the text there are some inconsistency in describing abbreviation i.e. FMD (line 95) is mentioned after phrase "Femoral Head Medialisation", later (line 124) FMD is written after phrase "Femoral Medialisation Value", finally in the line 296 FMD comes after phrase Femoral Head Diameter. It is suggested to correct those, for potential readers confusing abbreviation i.e. FMD. Limitations are listed correctly, but more specific explanation for practical purpose of this suggested ultrasound criteria in routine busy screening clinic should be given by authors.
Author Response
Manuscript ID: Diagnostics-3229276
Manuscript name
Assessing Femoral Head Medialization in Developmental Hip Dysplasia Type 1 and Type 2 Hip Separation
Dear Editor,
We appreciate the chance to submit a revised version of the manuscript. We value the time and effort you and the reviewers invested in offering feedback on our article and are thankful for the insightful remarks and significant enhancements to our work.Below is a detailed point-by-point response to the reviewers' criticisms and concerns.
Report 1:
Thank you for given me opportunity to review this interesting manuscript. From my perspective introduction is too long for readers who are not novice to this topic. It is suggested to describe the problem in brief manner and explain purposes of this study. In method section and further in the text there are some inconsistency in describing abbreviation i.e. FMD (line 95) is mentioned after phrase "Femoral Head Medialisation", later (line 124) FMD is written after phrase "Femoral Medialisation Value", finally in the line 296 FMD comes after phrase Femoral Head Diameter. It is suggested to correct those, for potential readers confusing abbreviation i.e. FMD. Limitations are listed correctly, but more specific explanation for practical purpose of this suggested ultrasound criteria in routine busy screening clinic should be given by authors.
Response:
-First of all, I am grateful for the dear referee's valuable comments and suggestions.
-In accordance with the referee's recommendations, the 'introduction' section has been articulated in a more descriptive fashion. The too complex paragraphs in the introduction have been streamlined and edited.
- The abbreviation 'FMD' in the text has been revised and employed with a more precise definition. The referee's employment of 'FMD' in lines 95 and 296 has been restructured using more descriptive and suitable terminology.
Thank you for your valuable comment. We have added the necessary explanation to our work.
“The FMD value described in our study can be performed by taking two measurements on a single ultrasound image during the same hip examination. This easy measurement method will eliminate the confusion in differentiating between type 1 and type 2 hips. Measurement in this respect can prevent unnecessary follow-up and misdiagnosis of patients. In this way, it can add practicality to intensive ultrasound examinations. “
Best wishes…
Onder Durmaz, MD
Reviewer 2 Report
Comments and Suggestions for Authors
This study is a new approach to the diagnosis of DDH by adding FMD as another diagnostic tool and thus opening new perspectives of investigation into a very important clinical problem. The introduction clearly points out the need to improve diagnostic techniques, and the research design is well-designed with the purpose of the present study. Methods were appropriate and fully presented reproducible steps of measurement of FMD; results were persuasive and supported an appropriate statistical analysis. On the contrary, the moderate AUC value would suggest a final discussion of the clinical implication and possible limitations, in particular, the moderate accuracy of the test and the relatively small sample size. The limitations-for example, no subclassification of the Type 2 hips-are slightly more elaborated to give weight to the manuscript. In general, the text is promising but requires some moderate editing to make it clear and readable. This study represents an important step forward in DDH diagnosis with potential to make a real difference clinically, pending verification in larger series.
Comments on the Quality of English LanguageModerate editing of English language required.
Author Response
Manuscript ID: Diagnostics-3229276
Manuscript name
Assessing Femoral Head Medialization in Developmental Hip Dysplasia Type 1 and Type 2 Hip Separation
Dear Editor,
We appreciate the chance to submit a revised version of the manuscript. We value the time and effort you and the reviewers invested in offering feedback on our article and are thankful for the insightful remarks and significant enhancements to our work.Below is a detailed point-by-point response to the reviewers' criticisms and concerns.
Report 2:
This study is a new approach to the diagnosis of DDH by adding FMD as another diagnostic tool and thus opening new perspectives of investigation into a very important clinical problem. The introduction clearly points out the need to improve diagnostic techniques, and the research design is well-designed with the purpose of the present study. Methods were appropriate and fully presented reproducible steps of measurement of FMD; results were persuasive and supported an appropriate statistical analysis. On the contrary, the moderate AUC value would suggest a final discussion of the clinical implication and possible limitations, in particular, the moderate accuracy of the test and the relatively small sample size. The limitations-for example, no subclassification of the Type 2 hips-are slightly more elaborated to give weight to the manuscript. In general, the text is promising but requires some moderate editing to make it clear and readable. This study represents an important step forward in DDH diagnosis with potential to make a real difference clinically, pending verification in larger series.
Response:
- I sincerely appreciate the esteemed referee's insightful comments and recommendations.
- In accordance with the recommendations of both referees, certain sentences in various paragraphs were changed for enhanced descriptiveness and incorporated into the main article. Minor inaccuracies in certain abbreviations have been rectified and incorporated into the appropriate lines in their correct format.
-At the suggestion of the referee, new additions and adjustments were made to the limitations regarding the AUC value and the non-subclassification of Type 2 hips.
Best wishes…
Onder Durmaz, MD